# Stochastic contextual bandits with graph feedback: from independence number to MAS number

**Yuxiao Wen**[†]    **Yanjun Han**[†]    **Zhengyuan Zhou**[†,*]

New York University[†]    Arena Technologies[*]

{yuxiaowen, yanjunhan}@nyu.edu   zz26@stern.nyu.edu

## Abstract

We consider contextual bandits with graph feedback, a class of interactive learning problems with richer structures than vanilla contextual bandits, where taking an action reveals the rewards for all neighboring actions in the feedback graph under all contexts. Unlike the multi-armed bandits setting where a growing literature has painted a near-complete understanding of graph feedback, much remains unexplored in the contextual bandits counterpart. In this paper, we make inroads into this inquiry by establishing a regret lower bound $\Omega(\sqrt{\beta_M(G)T})$, where $M$ is the number of contexts, $G$ is the feedback graph, and $\beta_M(G)$ is our proposed graph-theoretic quantity that characterizes the fundamental learning limit for this class of problems. Interestingly, $\beta_M(G)$ interpolates between $\alpha(G)$ (the independence number of the graph) and $\mathsf{m}(G)$ (the maximum acyclic subgraph (MAS) number of the graph) as the number of contexts $M$ varies. We also provide algorithms that achieve near-optimal regret for important classes of context sequences and/or feedback graphs, such as transitively closed graphs that find applications in auctions and inventory control. In particular, with many contexts, our results show that the MAS number essentially characterizes the statistical complexity for contextual bandits, as opposed to the independence number in multi-armed bandits.

## 1 Introduction

Contextual bandits encode a rich class of sequential decision making problems in reality, including clinical trials, personalized healthcare, dynamic pricing, recommendation systems (Bouneffouf et al., 2020). However, due to the exploration-exploitation trade-off and a potentially large context space, the pace of learning for contextual bandits could be slow, and the statistical complexity of learning could be costly for application scenarios with bandit feedback (Agarwal et al., 2012). There are two common approaches to alleviate the burden of sample complexity, either by exploiting the function class structure for the reward (Zhu and Mineiro, 2022), or by utilizing additional feedback available during exploration.

In this paper we focus on the second approach, and aim to exploit the feedback structure efficiently in contextual bandits. The framework of formulating the feedback structure as feedback graphs in bandits has a long history (Mannor and Shamir, 2011; Alon et al., 2015, 2017; Lykouris et al., 2020), where a direct edge between two actions indicates choosing one action provides the reward information for the other. Such settings have also been generalized to contextual cases (Balseiro et al., 2019; Dann et al., 2020; Han et al., 2024), where counterfactual rewards could be available under different contexts. Typical results in these settings are that, the statistical complexity of bandits with feedback graphs is characterized by some graph-theoretic quantities, such as the independence number or the maximum acyclic subgraph (MAS) number of the feedback graph.

To understand the influence of the presence of contexts on the statistical complexity of learning and to compare with multi-armed bandits, we focus on the *tabular* setting where the contexts are treated as

general variables determining the rewards. A widely studied alternative is the *structured* setting that leverages certain structures in the dependence on the context. Examples of the latter include linear contextual bandits (Auer, 2002; Agrawal and Goyal, 2013), which assume a linear reward function on the contexts, and their variants (Chu et al., 2011; Li et al., 2017; Agrawal and Devanur, 2016).

Despite the existing results, especially in multi-armed bandits where a near-complete characterization of the optimal regret is available (Alon et al., 2015; Kocák and Carpentier, 2023; Eldowa et al., 2024), the statistical complexity of contextual bandits with feedback graphs is much less understood. For example, consider the case where there is a feedback graph $G$ across the actions and a complete feedback graph across the contexts (termed as *complete cross-learning* in (Balseiro et al., 2019)). In this case, for a long time horizon $T$, the optimal regret scales as $\widetilde{\Theta}(\sqrt{\alpha(G)T})$ when there is only one context (Alon et al., 2015), but only an upper bound $\widetilde{O}(\sqrt{\mathsf{m}(G)T})$ is known regardless of the number of contexts (Dann et al., 2020). Here $\alpha(G)$ and $\mathsf{m}(G)$ denote the independence number and the MAS number of the graph $G$, respectively; we refer to Section 1.1 for the precise definitions. While $\alpha(G) = \mathsf{m}(G)$ for all undirected graphs, for directed graphs their gap could be significant. It is open if the change from $\alpha(G)$ to $\mathsf{m}(G)$ is essential with the increasing number of contexts, not to mention the precise dependence on the number of contexts.

## 1.1  Notations

For $n \in \mathbb{N}$, let $[n] := \{1, \cdots, n\}$. For two probability measures $P$ and $Q$ over the same space, let $\mathsf{TV}(P, Q) = \int |\mathrm{d}P - \mathrm{d}Q|/2$ be the total variation (TV) distance, and $\mathsf{KL}(P\|Q) = \int \mathrm{d}P \log(\mathrm{d}P/\mathrm{d}Q)$ be the Kullback-Leibler (KL) divergence. We use the standard asymptotic notations $O, \Omega, \Theta$, as well as $\widetilde{O}, \widetilde{\Omega}, \widetilde{\Theta}$ to denote respective meanings within polylogarithmic factors.

We use the following graph-theoretic notations. For a directed graph $G = (V, E)$, let $u \to v$ denote that $(u, v) \in E$. For $u \in V$, let $N_{\mathrm{out}}(u) = \{v \in V : u \to v\}$ be the set of out-neighbors of $u$ (including $u$ itself). We will also use $N_{\mathrm{out}}(A) = \cup_{v \in A} N_{\mathrm{out}}(v)$ to denote the set of all out-neighbors of vertices in $A$. The *independence number* $\alpha(G)$, *dominating number* $\delta(G)$, and *maximum acyclic subgraph (MAS) number* $\mathsf{m}(G)$ are defined as

$$\alpha(G) = \max\{|I| : I \subseteq V \text{ is an independent set, i.e. } u \not\to v, \forall u \neq v \in I\},$$
$$\delta(G) = \min\{|J| : J \subseteq V \text{ is a dominating set, i.e. } N_{\mathrm{out}}(J) = V\},$$
$$\mathsf{m}(G) = \max\{|D| : D \subseteq V \text{ induces an acyclic subgraph of } G\},$$

respectively. It is easy to show that $\max\{\alpha(G), \delta(G)\} \leq \mathsf{m}(G)$, with a possibly unbounded gap, and a probabilistic argument also shows that $\delta(G) = O(\alpha(G) \log |V|)$ (cf. Lemma A.1).

## 1.2  Our results

In this paper we focus on contextual bandits with both feedback graphs across actions and complete cross-learning across contexts. This setting was proposed in (Han et al., 2024), with applications to bidding in first-price auctions. As opposed to an arbitrary feedback graph across all context-action pairs in (Dann et al., 2020), we assume a complete cross-learning because of two reasons. First, in many scenarios the contexts encode different states which only play roles in the reward function; in other words, the counterfactual rewards for all contexts can be observed by plugging different contexts into the reward function. Such examples include bidding in auctions (Balseiro et al., 2019; Han et al., 2024) and sleeping bandits modeled in (Schneider and Zimmert, 2024). Second, this scenario is representative and sufficient to reflect the main ideas and findings of this paper. Discussion and results under more general settings is left to Section 4.1.

Throughout this paper we consider the following stochastic contextual bandits. At the beginning of each round $t \in [T]$ during the time horizon $T$, an oblivious adversary chooses a context $c_t \in [M]$ and reveals it to the learner, and the learner chooses an action $a_t \in [K]$. There is a strongly observable[1] directed feedback graph $G = ([K], E)$ across the actions such that all rewards in $(r_{t,c,a})_{c \in [M], (a_t, a) \in E}$ are observable, where we assume no structure in the rewards except that $r_{t,c,a} \in [0, 1]$. In our stochastic environment, the mean reward $\mathbb{E}[r_{t,c,a}] = \mu_{c,a}$ is unknown but invariant with time. We are interested in the characterization of the minimax regret achieved by the

---

[1]For every $a \in [K]$, either $a \to a$ or $a' \to a$ for all $a' \neq a$, as defined in (Alon et al., 2015).

learner:

$$\mathsf{R}_T^\star(G, M) = \inf_{\pi^T} \mathsf{R}_T(\pi^T; G, M)$$

$$= \inf_{\pi^T} \sup_{c^T} \sup_{\mu \in [0,1]^{K \times M}} \mathbb{E}\left[\sum_{t=1}^{T}\left(\max_{a^\star(c_t) \in [K]} \mu_{c_t, a^\star(c_t)} - \mu_{c_t, \pi_t(c_t)}\right)\right], \qquad (1)$$

where the infimum is over all admissible policies based on the available observations. In the sequel we might also constrain the class of context sequences to $c^T \in \mathcal{C}$, and we will use $\mathsf{R}_T^\star(G, M, \mathcal{C})$ and $\mathsf{R}_T(\pi^T; G, M, \mathcal{C})$ to denote the respective meanings by taking the supremum over $c^T \in \mathcal{C}$.

Our first result concerns a new lower bound on the minimax regret.

**Theorem 1.1** (Minimax lower bound). *For $T \geq \beta_M(G)^3$, it holds that $\mathsf{R}_T^\star(G, M) = \Omega(\sqrt{\beta_M(G)T})$, where the graph-theoretic quantity $\beta_M(G)$ is given by*

$$\beta_M(G) = \max\left\{\sum_{c=1}^{M} |I_c| : I_1, \cdots, I_M \text{ disjoint independent subsets of } [K], \text{ and } I_i \not\rightarrow I_j \text{ for } i < j\right\}, \qquad (2)$$

*and $I_i \not\rightarrow I_j$ means that $u \not\rightarrow v$ whenever $u \in I_i$ and $v \in I_j$.*

Theorem 1.1 provides a minimax lower bound on the optimal regret, depending on both the number of contexts $M$ and the feedback graph $G$. Note that the independent subsets $I_1, \ldots, I_M$ are allowed to be empty if needed. It is clear that $\beta_1(G) = \alpha(G)$ is the independence number, and $\beta_M(G) = \mathsf{m}(G)$ whenever $M \geq \mathsf{m}(G)$. This leads to the following corollary.

**Corollary 1.2** (Tightness of MAS number). *For any graph $G$, if $M \geq \mathsf{m}(G)$ and $T \geq \mathsf{m}(G)^3$, one has $\mathsf{R}_T^\star(G, M) = \Omega(\sqrt{\mathsf{m}(G)T})$.*

Corollary 1.2 shows that, the regret change from $\widetilde{\Theta}(\sqrt{\alpha(G)T})$ in multi-armed bandits to $\widetilde{O}(\sqrt{\mathsf{m}(G)T})$ in contextual bandits (Dann et al., 2020) is in fact not superfluous when there are many contexts. In other words, although the *independence number* determines the statistical complexity of multi-armed bandits with graph feedback, the statistical complexity in contextual bandits with many contexts is completely characterized by the *MAS number*.

For intermediate values of $M \in (1, \mathsf{m}(G))$, the next result shows that the quantity $\beta_M(G)$ is tight for a special class $\mathcal{C}_{\mathsf{SA}}$ of context sequence called *self-avoiding contexts*. A context sequence $(c_1, \cdots, c_T)$ is called self-avoiding iff $c_s = c_t$ for $s < t$ implies $c_s = c_{s+1} = \cdots = c_t$ (or in other words, contexts do not jump back). For example, 113222 is self-avoiding, but 12231 is not. This assumption is reasonable when contexts model a nonstationary environment changing slowly, e.g. the environment changes from season to season.

**Theorem 1.3** (Upper bound for self-avoiding contexts). *For self-avoiding contexts, there is a policy $\pi$ achieving $\mathsf{R}_T(\pi; G, M, \mathcal{C}_{\mathsf{SA}}) = \widetilde{O}(\sqrt{\beta_M(G)T})$. This policy can be implemented in polynomial-time, and does not need to know the context sequence in advance.*

As the minimax lower bound in Theorem 1.1 is actually shown under $\mathcal{C}_{\mathsf{SA}}$, for large $T$, Theorem 1.3 establishes a tight regret bound for stochastic contextual bandits with graph feedback and self-avoiding contexts. The policy used in Theorem 1.3 is based on arm elimination, where a central step of exploration is to solve a sequential game in general graphs which has minimax value $\widetilde{\Theta}(\beta_M(G))$ and could be of independent interest.

For general context sequences, we have a different sequential game in which we do not have a tight characterization of the minimax value in general. Instead, we have the following upper bound, which exhibits a gap compared with Theorem 1.1.

**Theorem 1.4** (Upper bound for general contexts). *For general contexts, there is a policy $\pi$ achieving*

$$\mathsf{R}_T(\pi; G, M) = \widetilde{O}\left(\sqrt{\min\left\{\overline{\beta}_M(G), \mathsf{m}(G)\right\}T}\right),$$

*where*

$$\overline{\beta}_M(G) = \max\left\{\sum_{c=1}^{M} |I_c| : I_1, \cdots, I_M \text{ are disjoint independent subsets of } [K]\right\}. \qquad (3)$$

Fortunately, additional assumptions on the feedback graph $G$ can be leveraged to recover the tight regret bound:

**Corollary 1.5** (Upper bound for transtively closed or undirected feedback). *For any undirected or transitively closed graph $G$, the policy $\pi$ in Theorem 1.4 achieves a near-optimal regret* $\mathsf{R}_T(\pi; G, M) = \widetilde{O}(\sqrt{\beta_M(G)T})$.

A directed graph $G$ is called *transitively closed* if $u \to v$ and $v \to w$ imply that $u \to w$. In reality directed feedback graphs are often transitively closed, for a directed structure of the feedback typically indicates a partial order over the actions. Examples include bidding in auctions (Zhao and Chen, 2019; Han et al., 2024) and inventory control (Huh and Rusmevichientong, 2009), both of which exhibit the one-sided feedback structure $i \to j$ for $i \leq j$. For general graphs, Theorem 1.4 gives another graph-theoretic quantity $\overline{\beta}_M(G)$. Note that $\overline{\beta}_M(G) \geq \beta_M(G)$ as there is no acyclic requirement between $I_c$'s in (3), which in turn is due to a technical difficulty of non-self-avoiding contexts. Further discussions on this gap are deferred to Section 4.3.

Interestingly, the upper bound quantities $\beta_M(G)$ and $\overline{\beta}_M(G)$ are not explicitly linear in $M$ and are always no larger than $\alpha(G)M$. Hence our results partially answer an open problem in (Hao et al., 2022, Remark 5.11) that if the dependence of regret bound $O(\sqrt{\alpha(G)MT})$ on $M$ can be improved.

## 1.3 Related work

The study of bandits with feedback graphs has a long history dating back to (Mannor and Shamir, 2011). For (both adversarial and stochastic) multi-armed bandits, a celebrated result in (Alon et al., 2015, 2017) shows that the optimal regret scales as $\widetilde{\Theta}(\sqrt{\alpha(G)T})$ if $T \geq \alpha(G)^3$; the case of smaller $T$ was settled in (Kocák and Carpentier, 2023), where the optimal regret is a mixture of $\sqrt{T}$ and $T^{2/3}$ rates. For stochastic bandits, simpler algorithms based on arm elimination or upper confidence bound (UCB) are also proposed (Lykouris et al., 2020; Han et al., 2024), while the UCB algorithm is only known to achieve an upper bound of $\widetilde{O}(\sqrt{\mathsf{m}(G)T})$.[2] In addition to strongly observable graphs we primarily focus on, weakly observable graphs have also drawn vast interest (Alon et al., 2015; Chen et al., 2021) where the optimal regret is characterized by the dominating number $\delta(G)$. There exploration plays a more significant role due to weaker observability of certain nodes, leading to an optimal regret $\widetilde{\Theta}(\delta(G)^{1/3}T^{2/3})$. We will briefly discuss the regret characterization of our contextual setting with weakly observable graphs in Section 4.1 and 4.2.

Recently, the graph feedback was also extended to contextual bandits under the name of "cross-learning" (Balseiro et al., 2019; Schneider and Zimmert, 2024). The work (Balseiro et al., 2019) considered both complete and partial cross-learning, and showed that the optimal regret for stochastic bandits with complete cross learning is $\widetilde{\Theta}(\sqrt{KT})$. Motivated by bidding in first-price auctions, (Han et al., 2024) generalized the setting to general graph feedback across actions and complete cross-learning across contexts, a setting used in the current paper. The finding in (Han et al., 2024) is that the effects of graph feedback and cross-learning could be "decoupled": a regret upper bound $\widetilde{O}(\sqrt{\min\{\alpha(G)M, K\}T})$ is shown, which is tight only for a special choice of the feedback graph $G$. The work (Dann et al., 2020) considered a tabular reinforcement learning setting with adversarial initial states, so that their setting with episode length $H = 1$ coincides with our problem with a general feedback graph $G$ across all context-action pairs. They showed that the UCB algorithm achieves a regret upper bound $\widetilde{O}(\sqrt{\mathsf{m}(G)T})$; however, their lower bound was only $\Omega(\sqrt{\alpha(G)T})$ when $T \geq \alpha(G)^3$. Therefore, tight lower bounds that work for general graphs $G$ are still underexplored in the literature, and our regret upper bounds in Theorems 1.3 and 1.4 also improve or generalize the existing results.

The problem of bandits with feedback is also closely related to partial monitoring games (Bartók et al., 2014). Although this is a more general setting which subsumes bandits with graph feedback, the results in the literature (Bartók et al., 2014; Lattimore, 2022; Foster et al., 2023a) typically have tight dependence on $T$, but often not on other parameters such as the dimensionality. Similar issues also applied to the recent line of work (Foster et al., 2021, 2023b) aiming to provide a unified complexity measure based on the decision-estimation coefficient (DEC); the nature of the two-point lower bound

---

[2]The result of (Lykouris et al., 2020) was stated using the independence number, but they only considered undirected graphs so that $\mathsf{m}(G) = \alpha(G)$.

used there often leaves a gap. We also point to some recent work (Zhang et al., 2024) which adopted the DEC framework and established regret bounds for contextual bandits with graph feedback, but no cross-learning across contexts, based on regression oracles.

## 2 Hard instance and the regret lower bound

In this section we sketch the proof of the minimax lower bound $R_T^\star(G, M, \mathcal{C}_{\mathsf{SA}}) = \Omega(\sqrt{\beta_M(G)T})$ for $T \geq \beta_M(G)^3$ and general $(G, M)$, implying Theorem 1.1. We first identify a hard instance that corresponds to the graph-theoretic quantity $\beta_M(G)$, and then present the core exploration-exploitation tradeoff in the proof to arrive at the fundamental limit of learning under this instance. This approach has been widely adopted in the bandit literature. The complete proof is deferred to Appendix B.

The proof uses the definition (2) of $\beta_M(G)$ to construct $M$ independent sets $I_1, \cdots, I_M$ such that $I_i \not\rightarrow I_j$ for $i < j$; by definition, the independent sets $I_1, \cdots, I_M$ are disjoint. We then construct a hard instance where the best action under context $c \in [M]$ is distributed uniformly over $I_c$; since $I_c$ is an independent set, this ensures that the learner must essentially explore all actions in $I_c$ under context $c$. Moreover, the context sequence $c^T$ is set to be $11 \cdots 122 \cdots 2 \cdots M$, i.e. never goes back to previous contexts. This order ensures that the exploration in $I_{c_1}$ during earlier rounds provides no information to the exploration in $I_{c_2}$ during later rounds, whenever $c_1 < c_2$. Naïvely, if the learner only explores in each $I_c$ under context $c$, then learning under each context $c$ becomes a multi-armed bandit problem (because $I_c$ is itself an independent set), and we can show lower bound $\sqrt{T \sum_{c=1}^M |I_c|}$ with appropriate context sequence $c^T$. Maximizing over all possible constructions gives the desired result.

It is possible, however, for the learner to choose actions outside $I_c$ to obtain information for the later rounds. To address this challenge, we use a delicate exploration-exploitation tradeoff argument to show that this pure exploration must incur a too large regret to be informative when $T \geq \beta_M(G)^3$. Specifically, consider the regret incurred by this pure exploration:

$$R_{\text{explore}} = \sum_{c=1}^M \sum_{t \in T_c} \mathbb{E}[\mathbb{1}(a_t \notin I_c)]$$

where $T_c = \{t \in [T] : c_t = c\}$. Then for some absolute constants $c_1$ and $c_2$, the tradeoff can be formulated as two lower bounds of the regret $R_T$:

$$R_T \geq c_1 \sqrt{\beta_M(G)T} \exp(-\beta_M(G)R_{\text{explore}}/T) \quad \text{and} \quad R_T \geq c_2 R_{\text{explore}}.$$

The first bound is decreasing in the amount of pure exploration, while the second one is increasing. Balancing this tradeoff gives the desired lower bound $R_T = \Omega\left(\sqrt{\beta_M(G)T}\right)$ for $T \geq \beta_M(G)^3$.

In summary, the key structure used in the proof is that $I_i \not\rightarrow I_j$ for $i < j$; we remark that this does not preclude the possibility that $I_j \rightarrow I_i$ for $j > i$, which underlies the change from $\alpha(G)$ to $\mathsf{m}(G)$ as the number of context increases.

## 3 Algorithms achieving the regret upper bounds

This section provides algorithms that achieve the claimed regret upper bounds in Theorems 1.3 and 1.4. The crux of these algorithms is to exploit the structure of the feedback graph and choose a small number of actions to explore. Depending on whether the context sequence is self-avoiding or not, the above problem can be reduced to two different kinds of sequential games on the feedback graph. Given solutions to the sequential games, Sections 3.2 and 3.3 will rely on the layering technique to use these solutions on each layer, and propose the final learning algorithms via arm elimination.

### 3.1 Two sequential games on graphs

In this section we introduce two sequential games on graphs which are purely combinatorial and independent of the learning process. We begin with the first sequential game.

**Definition 1** (Sequential game I). *Given a directed graph $G = (V, E)$ and a positive integer $M$, the sequential game consists of $M$ steps, where at each step $c = 1, \cdots, M$:*

1. *the adversary chooses a strongly observable subset $A_c \subseteq V$ disjoint from $N_{\text{out}}(\cup_{c' < c} D_{c'})$;*

2. *the learner chooses $D_c \subseteq A_c$ such that $D_c$ dominates $A_c$, i.e. $A_c \subseteq N_{\text{out}}(D_c)$.*[3]

*The learner's goal is to minimize the total size $\sum_{c=1}^{M} |D_c|$ of the sets $D_c$.*

The above sequential game is motivated by bandit learning under self-avoiding contexts. Consider a self-avoiding context sequence in the order of $1, 2, \cdots, M$. For $c \in [M]$, the set $A_c$ represents the "active set" of actions, i.e. the set of all probably good actions, yet to be explored when context $c$ first occurs. Thanks to the self-avoiding structure, "yet to be explored" means that $A_c$ must be disjoint from $N_{\text{out}}(\cup_{c' < c} D_{c'})$. The learner then plays a set of actions $D_c \subseteq A_c$ to ensure that all actions in $A_c$ have been explored at least once; we note that a good choice of $D_c$ not only aims to observe all of $A_c$, but also tries to observe as many actions as possible outside $A_c$ and make the complement of $N_{\text{out}}(\cup_{c' \leq c} D_{c'})$ small. The final cost $\sum_{c=1}^{M} |D_c|$ characterizes the overall sample complexity to explore every active action once over all contexts.

It is clear that the minimax value of this sequential game is given by

$$U_1^\star(G, M) = \max_{\substack{A_1 \subseteq V \\ A_1 \subseteq N_{\text{out}}(D_1)}} \min_{\substack{D_1 \subseteq A_1 \\ }} \cdots \max_{\substack{A_M \subseteq V \\ \cup_{c=1}^{M-1} D_c \not\supseteq A_M}} \min_{\substack{D_M \subseteq A_M \\ A_M \subseteq N_{\text{out}}(D_M)}} \sum_{c=1}^{M} |D_c|. \tag{4}$$

The following lemma characterizes the quantity $U_1^\star(G, M)$ up to an $O(\log |V|)$ factor.

**Lemma 3.1** (Minimax value of sequential game I). *There exists an absolute constant $C > 0$ that*

$$\beta_M(G) \leq U_1^\star(G, M) \leq C\beta_M(G) \log |V|.$$

*Moreover, the learner can achieve a slightly larger upper bound $O(\beta_M(G) \log^2 |V|)$ using a polynomial-time algorithm.*

The second sequential game is motivated by bandit learning with an arbitrary context sequence.

**Definition 2** (Sequential game II). *Given a directed graph $G = (V, E)$ and a positive integer $M$, the sequential game starts with an empty set $D_0 = \varnothing$, and at time $t = 1, 2, \cdots$:*

1. *the adversary chooses an integer $c_t \in [M]$ (and a set $A_{c_t} \subseteq V$ if $c_t$ does not appear before). The adversary must ensure that $A_{c_t} \backslash N_{\text{out}}(D_{t-1})$ is non-empty;*

2. *the learner picks a vertex $v_t \in A_{c_t}$ and updates $D_t \leftarrow D_{t-1} \cup \{v_t\}$.*

*The game terminates at time $T$ whenever the adversary has no further move (i.e. $\cup_c A_c \subseteq N_{\text{out}}(D_T)$), and the learner's goal is to minimize the duration $T$ of the game.*

The new sequential game reflects the case where the context sequence might not be self-avoiding, so instead of taking a set of actions at once, the learner now needs to take actions non-consecutively. Clearly the sequential game II is more difficult for the learner as it subsumes the sequential game I when the context sequence is self-avoiding: the set $D_c$ in Definition 1 is simply the collection of $v_t$'s in Definition 2 whenever $c_t = c$. Consequently, the minimax values satisfy $U_2^\star(G, M) \geq U_1^\star(G, M)$. The following lemma proves an upper bound on $U_2^\star(G, M)$.

**Lemma 3.2** (Minimax value of sequential game II). *There exists a polynomial-time algorithm for the learner which achieves*

$$U_2^\star(G, M) \leq \beta_{\text{dom}}(G, M) \leq \min\{\mathsf{m}(G), C\overline{\beta}_M(G) \log^2 |V|\},$$

*where $C > 0$ is an absolute constant, $\overline{\beta}_M(G)$ is given in (3), and*

$$\beta_{\text{dom}}(G, M) = \max \left\{ \sum_{c=1}^{M} |B_c| : \bigcup_c B_c \text{ is acyclic, } B_c \subseteq V_c \text{ dominates some } V_c \right.$$

$$\left. \text{with disjoint } V_1, \cdots, V_M \subseteq V, \text{ and } |B_c| \leq \delta(V_c)(1 + \log |V|). \right\}$$

---

[3]Both sets $(A_c, D_c)$ are allowed to be empty.

## 3.2 Learning under self-avoiding contexts

Given a learner's algorithm for the first sequential game, we are ready to provide an algorithm for bandit learning under any self-avoiding context sequence. The algorithm relies on the well-known idea of arm elimination (Even-Dar et al., 2006): for each context $c \in [M]$, we maintain an active set $A_c$ consisting of all probably good actions so far under this context based on usual confidence bounds of the rewards. To embed the sequential games into the algorithm, we further make use of the *layering technique* in (Lykouris et al., 2020; Dann et al., 2020): for $\ell \in \mathbb{N}$, we construct the set $A_{c,\ell}$ as the active set on layer $\ell$ such that all actions in $A_{c,\ell-1}$ have been taken for at least $\ell - 1$ times. In other words, the active set $A_{c,\ell}$ is formed based on $\ell - 1$ reward observations of all currently active actions. As higher layer indicates higher estimation accuracy, the learner now aims to minimize the duration of each layer $\ell$, which is precisely the place we will play an independent sequential game.

---

**Algorithm 1:** Arm elimination algorithm for self-avoiding contexts

---

**Input:** time horizon $T$, action set $[K]$, context set $[M]$, feedback graph $G$, a subroutine $\mathcal{A}$ for the sequential game I, failure probability $\delta \in (0, 1)$.
**Initialize:** active sets $A_{c,1} \leftarrow [K]$ for all contexts $c \in [M]$ on layer 1.
**for** $c = 1$ **to** $M$ **do**
    **for** $\ell = 1, 2, \cdots$ **do**
        compute $D_{c,\ell} \subseteq A_{c,\ell} \backslash N_{\text{out}}(\cup_{c'<c} D_{c',\ell})$ according to the subroutine $\mathcal{A}$, based on past plays

$$(A_{c',\ell} \backslash N_{\text{out}}(\cup_{i<c'} D_{i,\ell}))_{c' \leq c} \text{ and } (D_{c',\ell})_{c'<c};$$

        choose each action in $D_{c,\ell}$ once (break the loop if $c_t \neq c$ or $t > T$ during this process), and update $t$ accordingly;
        compute the empirical rewards $\bar{r}_{c,a}$ for all actions based on all historic reward observations;
        choose the following active set on the next layer:

$$A_{c,\ell+1} \leftarrow \left\{ a \in A_{c,\ell} : \bar{r}_{c,a} \geq \max_{a' \in A_{c,\ell}} \bar{r}_{c,a'} - 2\sqrt{\frac{\log(2MKT/\delta)}{\ell}} \right\}; \qquad (5)$$

        move to the next layer $\ell \leftarrow \ell + 1$.
    **end**
**end**

---

The description of the algorithm is summarized in Algorithm 1, and we assume without loss of generality that the self-avoiding contexts comes in the order of $1, \ldots, M$ (the duration of some contexts might be zero). During each context, Algorithm 1 sequentially constructs a shrinking sequence of active sets $A_{c,1} \supseteq A_{c,2} \supseteq \cdots$, and on each layer $\ell$, the algorithm plays the sequential game I based on the current status (past plays $(A_{c',\ell})_{c' \leq c}$, or equivalently $(A_{c',\ell} \backslash N_{\text{out}}(\cup_{i<c'} D_{i,\ell}))_{c' \leq c}$, of the adversary, and past plays $(D_{c',\ell})_{c'<c}$ of the learner).[4] After the rewards of all actions of $A_{c,\ell}$ have been observed once, the algorithm constructs the active set $A_{c,\ell+1}$ for the next layer based on the confidence bound (5) and sample size $\ell$.

The following theorem summarizes the performance of the algorithm.

**Theorem 3.3** (Regret upper bound of Algorithm 1). *Let the subroutine $\mathcal{A}$ for the sequential game I be the polynomial-time algorithm given by Lemma 3.1. Then with probability at least $1 - \delta$, the regret of Algorithm 1 is upper bounded by*

$$\mathsf{R}_T(\text{Alg 1}; G, M, \mathcal{C}_{\mathsf{SA}}) = O\left( \sqrt{T \beta_M(G) \log^2(K) \log(MKT/\delta)} \right).$$

On a high level, by classical confidence bound arguments, each action chosen on layer $\ell$ suffers from an instantaneous regret $\widetilde{O}(1/\sqrt{\ell})$. Moreover, Lemma 3.1 shows that the number of actions chosen on a given layer is at most $\widetilde{O}(\beta_M(G))$. A combination of these two observations leads to the $\widetilde{O}(\sqrt{\beta_M(G)T})$ upper bound in Theorem 3.3, and a full proof is provided in Appendix C.

---

[4]It is possible that, at some layer $\ell$ and for some context $c$, every action active under $c$ has been explored, i.e. $A_{c,\ell} \backslash N_{\text{out}}(\cup_{c'<c} D_{i,\ell}) = \varnothing$. In this case, the learner simply skips to next layers by choosing $D_c = \varnothing$.

### 3.3 Learning under general contexts

The learning algorithm under a general context sequence is described in Algorithm 2. Similar to Algorithm 1, for each context $c$ we break the learning process into different layers, construct active sets $A_{c,\ell}$ for each layer, and move to the next layer whenever all actions in $A_{c,\ell}$ have been observed once on layer $\ell$. The only difference lies in the choice of actions on layer $\ell$, where the plays from the sequential game II are now used. The following theorem summarizes the performance of Algorithm 2, whose proof is very similar to Theorem 3.3 and deferred to Appendix C.

---

**Algorithm 2:** Arm elimination under general contexts

---

**Input:** time horizon $T$, action set $[K]$, context set $[M]$, feedback graph $G$, a subroutine $\mathcal{A}$ for the sequential game II, failure probability $\delta \in (0, 1)$.
**Initialize:** active sets $A_{c,\ell} \leftarrow [K]$ for all contexts $c \in [M]$ and layers $\ell \geq 1$; set of actions $D_\ell \leftarrow \varnothing$ chosen on layer $\ell$; the current layer index $\ell(c) \leftarrow 1$ for all $c \in [M]$.
**for** $t = 1$ **to** $T$ **do**
    receive the context $c_t$, and compute the current layer index $\ell_t = \ell(c_t)$;
    according to subroutine $\mathcal{A}$, choose an action $a_t \in A_{c_t,\ell_t}$ based on the active sets
    $(A_{c,\ell_t})_{c \in [M]}$ and previously taken actions $D_{\ell_t}$ on the current layer;
    update the set of actions on layer $\ell_t$ via $D_{\ell_t} \leftarrow D_{\ell_t} \cup \{a_t\}$;
    **for** $c \in [M]$ **do**
        compute the new layer index $\ell_{\mathsf{new}}(c) = \min\{\ell : A_{c,\ell} \nsubseteq N_{\mathsf{out}}(D_\ell)\}$;
        **if** $\ell_{\mathsf{new}}(c) > \ell(c)$ **then**
            compute the empirical rewards $\bar{r}_{c,a}$ for all actions based on all historic observations;
            choose the following active set on the new layer:

$$A_{c,\ell_{\mathsf{new}}(c)} \leftarrow \left\{ a \in A_{c,\ell(c)} : \bar{r}_{c,a} \geq \max_{a' \in A_{c,\ell(c)}} \bar{r}_{c,a'} - 2\sqrt{\frac{\log(2MKT/\delta)}{\ell_{\mathsf{new}}(c) - 1}} \right\};$$

            update the layer index $\ell(c) \leftarrow \ell_{\mathsf{new}}(c)$.
        **end**
    **end**
**end**

---

**Theorem 3.4** (Regret upper bound of Algorithm 2). *Let the subroutine $\mathcal{A}$ for the sequential game II be the polynomial-time algorithm given by Lemma 3.2. Then with probability at least $1 - \delta$, the regret of Algorithm 2 is upper bounded by*

$$\mathsf{R}_T(\mathsf{Alg}\ 2; G, M) = O\left( \sqrt{T\beta_{\mathsf{dom}}(G, M) \log(MKT/\delta)} \right).$$

By the second inequality in Lemma 3.2, Theorem 3.4 implies Theorem 1.4. Corollary 1.5 then follows from the following result.

**Lemma 3.5.** *For undirected or transitively closed graph $G$, it holds that $\beta_{\mathsf{dom}}(G, M) = O(\beta_M(G) \log |V|)$.*

## 4 Discussions

### 4.1 Weakly observable feedback graphs

Naturally, we may ask what results we would get under a weaker assumption / a more general feedback structure. If the feedback graph $G$ is instead weakly observable[5], then under complete cross-learning, an explore-then-commit (ETC) policy can achieve regret $\widetilde{O}(\delta(G)^{1/3}T^{2/3})$ by first exploring the minimum dominating set[6] uniformly for time $\delta(G)^{1/3}T^{2/3}$, and then committing to the empirically best action that has suboptimality bounded by $\widetilde{O}(\delta(G)^{1/3}T^{-1/3})$ with high probability. This matches the existing lower bound in (Alon et al., 2015) and is hence near-optimal.

---

[5]In the language of (Alon et al., 2015), a graph $G$ is weakly observable if $N_{\mathsf{in}}(a) \neq \varnothing$ for all $a \in [K]$, and there exists $a_0 \in [K]$ such that $\{a_0\}, [K] \backslash \{a_0\} \nsubseteq N_{\mathsf{in}}(a_0)$.
[6]Note that a $(1 + \log K)$-approximate minimum dominating set can be found efficiently by Lemma A.2.

## 4.2 Incomplete cross-learning

It is possible to further relax the assumption of complete cross-learning. Suppose the feedback across contexts is characterized by another directed graph $G_{[M]}$ (and denote $G_{[K]}$ across actions respectively), and consider a product feedback graph $G_{[K]} \times G_{[M]}$ over the context-action pairs such that $(a_1, c_1) \to (a_2, c_2)$ if $a_1 \to a_2$ in $G_{[K]}$ and $c_1 \to c_2$ in $G_{[M]}$. Then we can get the following generalized results.

### 4.2.1 Weakly observable feedback graphs on actions

When the feedback graph $G_{[K]}$ is weakly observable, following the argument in Section 4.1, we can achieve regret $\widetilde{O}\left(\left(\delta(G_{[K]})\mathsf{m}(G_{[M]})\right)^{1/3}T^{2/3}\right)$ by running an ETC subroutine for each context as follows: for every context $c \in [M]$, we keep an "exploration" counter $n_c$. At each time $t$ with context $c_t$, if $n_{c_t} \lesssim \delta(G_{[K]})^{1/3}\mathsf{m}(G_{[M]})^{-2/3}T^{2/3}$, we are in the "exploration" stage and continue to uniformly explore the minimum dominating set of $G_{[K]}$. Then we increase the counter for all observed contexts, i.e. $n_c \leftarrow n_c + 1$ for all $c \in N_{\text{out}}(c_t)$ in $G_{[M]}$. Otherwise, we "commit" to the empirically best action that has suboptimality bounded by $\widetilde{O}\left(\left(\delta(G_{[K]})\mathsf{m}(G_{[M]})\right)^{1/3}T^{-1/3}\right)$ with high probability.

The key observation is that the number of times we are in the "exploration" stage is $\widetilde{O}\left(\left(\delta(G_{[K]})\mathsf{m}(G_{[M]})\right)^{1/3}T^{2/3}\right)$. This can be seen from a layering argument, similar to the one in Section 3.2, that the number of actually played contexts on each layer is at most $\mathsf{m}(G_{[M]})$. Together with the bounded rewards and the bounded suboptimality in the "commit" stage, this proves the regret upper bound.

Combining the context sequence construction in Section 2 and the lower bound argument in (Alon et al., 2015), one can also prove a matching lower bound $\Omega\left(\left(\delta(G_{[K]})\mathsf{m}(G_{[M]})\right)^{1/3}T^{2/3}\right)$.

### 4.2.2 Strongly observable feedback graphs on actions

When $G_{[K]}$ is strongly observable, it is straightforward to generalize our upper (for self-avoiding contexts) and lower bounds in Theorem 1.1 and 1.3 with $\beta_M(G_{[K]})$ replaced by

$$\beta_M(G_{[K]} \times G_{[M]}) = \max\left\{\sum_{c=1}^{M} |I_c| : I_c \text{ independent subset of } [K] \times \{c\}, \text{ and } I_c \not\to I_{c'} \text{ for } c < c'\right\}$$

and $\beta_{\mathsf{dom}}(G_{[K]})$ in Theorem 3.4 by

$$\beta_{\mathsf{dom}}(G_{[K]} \times G_{[M]}) = \max\left\{\sum_{c=1}^{M} |B_c| : \bigcup_c B_c \text{ is acyclic in } G_{[K]} \times G_{[M]},\right.$$

$$\left. B_c \text{ is a } (1 + \log K)\text{-approx min dominating set of some subsets } V_c \subseteq G_{[K]} \times \{c\}\right\}$$

where the new graph quantities are defined on the product graph $G_{[K]} \times G_{[M]}$. For general contexts, this gives a tight upper bound $\widetilde{O}\left(\sqrt{T\beta_M(G_{[K]} \times G_{[M]})}\right)$ when $G_{[K]}$ and $G_{[M]}$ are either both *undirected* or both *transitively closed*[7]. Most generally, we have a loose upper bound $\widetilde{O}\left(\sqrt{T\min\{\mathsf{m}(G_{[K]} \times G_{[M]}), \alpha(G_{[K]})M\}}\right)$.

## 4.3 Gap between upper and lower bounds

Although we provide tight upper and lower bounds for specific classes of context sequences (self-avoiding in Theorem 1.3) or feedback graphs (undirected or transitively closed in Corollary 1.5), in general the quantities $\beta_M(G)$ in Theorem 1.1 and $\min\{\overline{\beta}_M(G), \mathsf{m}(G)\}$ in Theorem 1.4 exhibit a gap. The following lemma gives an upper bound on this gap.

---

[7]We prove this statement in Appendix C.7 due to space limit.

**Lemma 4.1.** *For any graph $G$, it holds that*

$$\beta_M(G) \leq \min\{\overline{\beta}_M(G), \mathsf{m}(G)\} \leq \max\left\{\frac{\rho(G)}{M}, 1\right\}\beta_M(G),$$

*where $\rho(G)$ denotes the length of the longest path in $G$.*

Lemma 4.1 shows that if $G$ does not contain long paths or $M$ is large, the gap between $\overline{\beta}_M(G)$ and $\beta_M(G)$ is not significant. We also comment on the challenge of closing this gap. First, we do not know a tight characterization of the minimax value of the sequential game II (cf. Definition 2), and the upper bound $\beta_{\mathsf{dom}}(G, M)$ in Lemma 3.2 could be loose, as shown in the following example.

**Example 1.** *Consider an acyclic graph $G = (V, E)$ with $KM$ vertices $\{(i, j)\}_{i \in [K], j \in [M]}$ and edges $(i, j) \to (i', j')$ if either $i < i'$ and $j \neq j'$, or $i = i'$ and $j < j'$, and all self-loops. By choosing $B_c = V_c = \{(i, c) : i \in [K]\}$ in the definition of $\beta_{\mathsf{dom}}(G, M)$ in Lemma 3.2, it is clear that $\beta_{\mathsf{dom}}(G, M) = KM$. However, we show that the minimax value is $U_2^\star(G, M) = K + M - 1$. The lower bound follows from $U_2^\star(G, M) \geq U_1^\star(G, M) \geq \beta_M(G) \geq K + M - 1$, as $I_1 = \{(i, M) : i \in [K]\}$ and $I_c = \{(1, M + 1 - c)\}$ for $2 \leq c \leq M$ satisfy the constraints in the definition of $\beta_M(G)$ in (2). For the upper bound, we consider the following strategy for the learner in the sequential game II: $v_t = (i_t, j_t)$ is the smallest element (under the lexicographic order over pairs) in $A_{c_t} \setminus N_{\mathsf{out}}(D_{t-1})$. To show why $U_2^\star(G, M) \leq K + M - 1$, let $D_c$ be the final set of vertices chosen by the learner under context $c$. By the lexicographic order and the structure of $G$, each $D_c$ can only consist of vertices in one column. Moreover, for different $c \neq c'$, the row indices of $D_c$ must be entirely no smaller or entirely no larger than the row indices of $D_{c'}$. These constraints ensure that $\sum_{c=1}^{M} |D_c| \leq K + M - 1$.*

*This example shows the importance of non-greedy approaches when choosing $v_t$. In the special case where $A_c = \{(i, c) : i \in [K]\}$ is the $c$-th column, within $A_c$ this is an independent set, so any greedy approach that does not look outside $A_c$ will treat the vertices in $A_c$ indifferently. In contrast, the above approach makes use of the global structure of the graph $G$.*

The second challenge lies in the proof of the lower bound. Instead of the sequential game where the adversary and the learner take turns to play actions, the current lower bound argument assumes that the adversary tells all his plays to the learner ahead of time. We expect the sequential structure to be equally important for the lower bounds, and it is interesting to work out an argument for the minimax lower bound to arrive at a sequential quantity like $U_2^\star(G, M)$.

### 4.4 Other open problems

**Performance of the UCB algorithm.** The UCB algorithm under feedback graphs has been analyzed for both multi-armed (Lykouris et al., 2020) and contextual bandits (Dann et al., 2020). However, both results only show a regret upper bound $\widetilde{O}(\sqrt{\mathsf{m}(G)T})$, even in the case of multi-armed bandits (i.e. $M = 1$). It is interesting to understand for algorithms without forced exploration (such as UCB), if the MAS number $\mathsf{m}(G)$ (rather than $\alpha(G)$ or $\beta_M(G)$) turns out to be fundamental.

**Regret for small $T$.** Note that our upper bounds hold for all values of $T$, but our lower bound requires $T \geq \beta_M(G)^3$. This is not an artifact of the analysis, as the optimal regret becomes fundamentally different for smaller $T$. The case of multi-armed bandits has been solved completely in a recent work (Kocák and Carpentier, 2023), where the regret is a mixture of $\sqrt{T}$ and $T^{2/3}$ terms. We anticipate the same behavior for contextual bandits, but the exact form is unknown.

**Stochastic contexts.** In this paper we assume that the contexts are generated adversarially, but the case of stochastic contexts also draws some recent attention (Balseiro et al., 2019; Schneider and Zimmert, 2024), and sometimes there is a fundamental gap between the optimal performances under stochastic and adversarial contexts (Han et al., 2024). It is an interesting question whether this is the case for contextual bandits with graph feedback.

## Acknowledgement

This work is generously funded by the NSF grant CCF 2106508.

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

## A Auxilary Lemmas

**Lemma A.1** (Lemma 8 of (Alon et al., 2015)). *For any directed graph $G = (V, E)$, one has $\delta(G) \le 50\alpha(G)\log|V|$.*

For a directed graph $G$, there is a well-known approximate algorithm for finding the smallest dominating set: starting from $D = \varnothing$, recursively find the vertex $v$ with the maximum out-degree in the subgraph induced by $V\backslash N_{\text{out}}(D)$, and update $D \leftarrow D \cup \{v\}$. The following lemma summarizes the performance of this algorithm.

**Lemma A.2** (Chvatal (1979)). *For any graph $G = (V, E)$, the above procedure outputs a dominating set $D$ with*
$$|D| \le (1 + \log|V|)\delta(G).$$

**Lemma A.3** (A special case of Lemma 3 in (Gao et al., 2019)). *Let $Q_1, \ldots Q_n$ be probability measures on some common measure space $(\Omega, \mathcal{F})$, with $n \ge 2$, and $\Phi : \Omega \to [n]$ any measurable test function. Then*
$$\frac{1}{n}\sum_{i=1}^{n} Q_i(\Phi \ne i) \ge \frac{1}{2n}\sum_{i=2}^{n}\exp(-\mathsf{KL}(Q_i\|Q_1)).$$

## B Deferred Proof for the Lower Bound

In this appendix, we give the complete proof of the minimax lower bound $\mathsf{R}^\star_T(G, M, \mathcal{C}_{\mathsf{SA}}) = \Omega(\sqrt{\beta_M(G)T})$ for $T \ge \beta_M(G)^3$ and general $(G, M)$, implying Theorem 1.1.

Let $I_1, \cdots, I_M$ be the independent sets achieving the maximum in (2), by removing empty sets, combining $(I_i, I_{i+1})$ whenever $|I_i| = 1$, and possibly removing the last set $I_M$ if $|I_M| = 1$, we arrive at disjoint subsets $J_1, \cdots, J_m$ of $[K]$ such that the following conditions hold:

- $m \le M$, $K_c \triangleq |J_c| \ge 2$ for all $c \in [m]$, and $J_i \nrightarrow J_j$ for $i < j$;
- the only possible non-self-loop edges among $J_c = \{a_{c,1}, \cdots, a_{c,K_c}\}$ can only point to $a_{c,1}$;
- $\sum_{c=1}^{m} K_c \ge \sum_{c=1}^{M} |I_c| - 1 = \beta_M(G) - 1 \ge \beta_M(G)/2$ whenever $\beta_M(G) \ge 2$.[8]

Given sets $J_1, \cdots, J_m$, we are ready to specify the hard instance. Let $u = (u_1, \cdots, u_m) \in \Omega := [K_1] \times \cdots \times [K_m]$ be a parameter vector, the joint reward distribution $P^u$ of $(r_{t,c,a})_{c\in[M],a\in[K]}$ is a product distribution $P^u = \prod_{c\in[M],a\in[K]} \mathsf{Bern}(\mu^u_{c,a})$, where the mean parameters for the Bernoulli distributions are $\mu^u_{c,a} = 0$ whenever $c > m$, and

$$\mu^u_{c,a} = \begin{cases} \frac{1}{4} + \Delta & \text{if } a = a_{c,1}, \\ \frac{1}{4} + 2\Delta & \text{if } a = a_{c,u_c} \text{ and } u_c \ne 1, \\ \frac{1}{4} & \text{if } a \in J_c\backslash\{a_{c,u_c}\}, \\ 0 & \text{if } a \notin J_c, \end{cases} \quad \text{for } c \in [m].$$

Here $\Delta \in (0, 1/4)$ is a gap parameter to be chosen later. We summarize some useful properties from the above construction:

1. under context $c \in [m]$, the best action under $P_u$ is $a_{c,u_c}$, and all other actions suffer from an instantaneous regret at least $\Delta$;

2. under context $c \in [m]$, actions outside $J_c$ suffer from an instantaneous regret at least $1/4$;

3. for $u = (u_1, \cdots, u_m) \in \Omega$ and $u^c := (u_1, \cdots, u_{c-1}, 1, u_{c+1}, \cdots, u_m)$, the KL divergence between the observed reward distributions $P^u(a)$ and $P^{u^c}(a)$ when choosing the action $a$ is

$$\mathsf{KL}(P^{u^c}(a)\|P^u(a)) \overset{(a)}{=} \begin{cases} \mathsf{KL}(\mathsf{Bern}(1/4)\|\mathsf{Bern}(1/4 + 2\Delta)) & \text{if } a \to a_{c,u_c} \\ 0 & \text{otherwise} \end{cases}$$
$$\overset{(b)}{\le} \frac{64\Delta^2}{3}\mathbb{1}(a \notin J_{\le c}\backslash\{a_{c,u_c}\}).$$

---

[8] When $\beta_M(G) = 1$, the $\Omega(\sqrt{T})$ regret lower bound is trivially true even under full information feedback and $M = 1$.

Here (a) follows from our construction of $P^u$ that the only difference between $P^u$ and $P^{u^c}$ is the reward of action $a_{c,u_c}$, which is observed iff $a \to a_{c,u_c}$; (b) is due to the property of $(J_1, \cdots, J_m)$ that any action in $J_{\leq c} \backslash \{a_{c,u_c}\}$ does not point to $a_{c,u_c}$, where $J_{\leq c} := \cup_{c' \leq c} J_{c'}$.

Finally, we partition the time horizon $[T]$ into consecutive blocks $T_1, \cdots, T_m$ (whose sizes will be specified later), and choose the context sequence as $c_t = c$ for all $t \in T_c$. For a fixed policy, let $\mathsf{R}_T$ be the worst-case expected regret of this policy. By the second property of the construction, it is clear that for all $u \in \Omega$,

$$\mathsf{R}_T \geq \frac{1}{4} \sum_{c=1}^{m} \sum_{t \in T_c} \mathbb{E}_{(P^u)^{\otimes(t-1)}}[\mathbb{1}(a_t \notin J_c)]. \tag{6}$$

When $u$ is uniformly distributed over $\Omega$, we also have

$$\mathsf{R}_T \overset{(a)}{\geq} \mathbb{E}_u \left[ \Delta \sum_{c=1}^{m} \sum_{t \in T_c} \mathbb{E}_{(P^u)^{\otimes(t-1)}}[\mathbb{1}(a_t \neq a_{c,u_c})] \right]$$

$$\overset{(b)}{=} \Delta \sum_{c=1}^{m} \sum_{t \in T_c} \mathbb{E}_{u \backslash \{u_c\}} \left[ \mathbb{E}_{u_c} \left[ \mathbb{E}_{(P^u)^{\otimes(t-1)}}[\mathbb{1}(a_t \neq a_{c,u_c})] \right] \right]$$

$$\overset{(c)}{\geq} \Delta \sum_{c=1}^{m} \sum_{t \in T_c} \mathbb{E}_{u \backslash \{u_c\}} \left[ \frac{1}{2K_c} \sum_{u_c=2}^{K_c} \exp\left( -\mathsf{KL}\left( (P^{u^c})^{\otimes(t-1)} \big\| (P^u)^{\otimes(t-1)} \right) \right) \right] \tag{7}$$

$$\overset{(d)}{\geq} \Delta \sum_{c=1}^{m} \sum_{t \in T_c} \mathbb{E}_{u \backslash \{u_c\}} \left[ \frac{K_c - 1}{2K_c} \exp\left( -\frac{1}{K_c - 1} \sum_{u_c=2}^{K_c} \mathsf{KL}\left( (P^{u^c})^{\otimes(t-1)} \big\| (P^u)^{\otimes(t-1)} \right) \right) \right]$$

$$\overset{(e)}{\geq} \frac{\Delta}{4} \sum_{c=1}^{m} \sum_{t \in T_c} \mathbb{E}_{u \backslash \{u_c\}} \left[ \exp\left( -\frac{64\Delta^2}{3(K_c - 1)} \sum_{u_c=2}^{K_c} \sum_{s<t} \mathbb{E}_{(P^{u^c})^{\otimes(s-1)}}[\mathbb{1}(a_s \notin J_{\leq c} \backslash \{a_{c,u_c}\})] \right) \right],$$

where (a) lower bounds the minimax regret by the Bayes regret, with the help of the first property; (b) decomposes the expectation over uniformly distributed $u$ into $u \backslash \{u_c\}$ and $u_c \in [K_c]$; (c) follows from Lemma A.3; (d) uses the convexity of $x \mapsto e^{-x}$; (e) results from the chain rule of KL divergence, the third property of the construction, and that $K_c \geq 2$ for all $c \in [m]$.

Next we upper bound the exponent in (7) as

$$\sum_{u_c=2}^{K_c} \sum_{s<t} \mathbb{E}_{(P^{u^c})^{\otimes(s-1)}}[\mathbb{1}(a_s \notin J_{\leq c} \backslash \{a_{c,u_c}\})]$$

$$\leq \sum_{u_c=2}^{K_c} \sum_{c'<c} \sum_{s \in T_{c'}} \mathbb{E}_{(P^{u^c})^{\otimes(s-1)}}[\mathbb{1}(a_s \notin J_{c'})] + \sum_{u_c=2}^{K_c} \sum_{\substack{s \in T_c \\ s<t}} \mathbb{E}_{(P^{u^c})^{\otimes(s-1)}}[\mathbb{1}(a_s \notin J_{\leq c}) + \mathbb{1}(a_s = a_{c,u_c})]$$

$$\leq \sum_{u_c=2}^{K_c} \sum_{c' \leq c} \sum_{s \in T_{c'}} \mathbb{E}_{(P^{u^c})^{\otimes(s-1)}}[\mathbb{1}(a_s \notin J_{c'})] + \sum_{u_c=2}^{K_c} \sum_{s \in T_c} \mathbb{E}_{(P^{u^c})^{\otimes(s-1)}}[\mathbb{1}(a_s = a_{c,u_c})]$$

$$\overset{(6)}{\leq} 4(K_c - 1)\mathsf{R}_T + \sum_{u_c=2}^{K_c} \sum_{s \in T_c} \mathbb{E}_{(P^{u^c})^{\otimes(s-1)}}[\mathbb{1}(a_s = a_{c,u_c})].$$

Plugging it back into (7), we get

$$\mathsf{R}_T \geq \frac{\Delta}{4} \sum_{c=1}^{m} \sum_{t \in T_c} \mathbb{E}_{u \backslash \{u_c\}} \left[ \exp\left( -\frac{64\Delta^2}{3(K_c - 1)} \left( 4(K_c - 1)\mathsf{R}_T + \sum_{u_c=2}^{K_c} \sum_{s \in T_c} \mathbb{E}_{(P^{u^c})^{\otimes(s-1)}}[\mathbb{1}(a_s = a_{c,u_c})] \right) \right) \right]$$

$$\overset{(f)}{\geq} \frac{\Delta}{4} \sum_{c=1}^{M} \sum_{t \in T_c} \mathbb{E}_{u \backslash \{u_c\}} \left[ \exp\left( -\frac{64\Delta^2}{3} \left( 4\mathsf{R}_T + \frac{|T_c|}{K_c - 1} \right) \right) \right],$$

where (f) crucially uses that $u^c = (u_1, \cdots, u_{c-1}, 1, u_{c+1}, \cdots, u_m)$ does not depend on $u_c$, so that the sum may be moved inside the expectation to get $\sum_{u_c=2}^{K_c} \mathbb{1}(a_s = a_{c,u_c}) \leq 1$. Now choosing

$$|T_c| = \frac{K_c}{\sum_{c'=1}^m K_{c'}} \cdot T \leq \frac{2K_c T}{\beta_M(G)}, \qquad \Delta = \sqrt{\frac{\beta_M(G)}{16T}} \in \left(0, \frac{1}{4}\right),$$

we arrive at the final lower bound

$$\mathsf{R}_T \geq \frac{\sqrt{\beta_M(G)T}}{16} \exp\left(-\frac{4\beta_M(G)}{3T}\left(4\mathsf{R}_T + \frac{4T}{\beta_M(G)}\right)\right)$$

$$\geq \frac{\sqrt{\beta_M(G)T}}{16e^6} \exp\left(-\frac{16\beta_M(G)}{3T}\mathsf{R}_T\right) \geq \frac{\sqrt{\beta_M(G)T}}{16e^6} \exp\left(-\frac{16\mathsf{R}_T}{3\sqrt{\beta_M(G)T}}\right), \qquad (8)$$

where the last inequality is due to the assumption $T \geq \beta_M(G)^3$. Now we readily conclude from (8) the desired lower bound $\mathsf{R}_T = \Omega(\sqrt{\beta_M(G)T})$.

## C Deferred Proofs for the Upper Bounds

Throughout the proofs, we will use $\alpha(A) \triangleq \alpha(G|_A)$ (resp. $\delta(A)$, $\mathsf{m}(A)$) to denote the independence number (resp. dominating number, MAS number) of the subgraph induced by $A \subseteq V$ when the graph $G$ is clear from the context.

### C.1 Proof of Lemma 3.1

The lower bound $U_1^\star(G, M) \geq \beta_M(G)$ is easy: let $I_1, \cdots, I_M$ be $M$ independent sets with $I_i \not\to I_j$ for all $i < j$. Then the choice $A_c = I_c$ is always feasible for the adversary, for $I_c$ is disjoint from $N_{\text{out}}(\cup_{c'<c}I_{c'})$. For the learner, the only subset $D_c \subseteq I_c$ which dominates $I_c$ is $D_c = I_c$, hence $U_1^\star(G, M) \geq \sum_{c=1}^M |I_c|$. Taking the maximum then gives $U_1^\star(G, M) \geq \beta_M(G)$ by (2).

To prove the upper bound $U_1^\star(G, M) \leq C\beta_M(G) \log |V|$, the learner chooses $D_c$ as follows. Given $A_c$, the learner finds the smallest dominating set $J_c \subseteq A_c$ and the largest independent set $I_c \subseteq A_c$, and sets $D_c = I_c \cup J_c$. Clearly this choice of $D_c$ is feasible for the learner, and since $I_i \not\to A_j$ for $i < j$, we have $I_i \not\to I_j$ as well. Consequently,

$$U^\star(G, M) \leq \sum_{c=1}^M |D_c| \leq \sum_{c=1}^M (|J_c| + |I_c|) \overset{(a)}{=} \sum_{c=1}^M O(|I_c| \log |V|) \overset{(b)}{=} O(\beta_M(G) \log |V|),$$

where (a) uses $|J_c| = \delta(A_c) = O(\alpha(A_c) \log |V|) = O(|I_c| \log |V|)$ in Lemma A.1, and (b) follows from the definition of $\beta_M(G)$ in (2).

Since it is NP-hard to find either the smallest dominating set or the largest independent set (Karp, 2010; Grandoni, 2006), the above choice of $D_c$ is not computationally efficient. To arrive at a polynomial-time algorithm, we may use a greedy algorithm to find an $O(\log |V|)$-approximate smallest dominating set $\widetilde{J}_c$ such that $|\widetilde{J}_c| = O(\delta(A_c) \log |V|)$ (cf. Lemma A.2). For $I_c$, although finding the largest independent set is APX-hard (Feige et al., 1991), the constructive proof of (Alon et al., 2015, Lemma 8) gives a polynomial-time randomized algorithm which finds $\widetilde{I}_c \subseteq A_c$ such that $|\widetilde{I}_c| = \Omega(\delta(A_c)/\log |V|)$ and the average degree among $\widetilde{I}_c$ is at most $O(1)$. The learner now chooses $D_c = \widetilde{I}_c \cup \widetilde{J}_c$. By the average degree constraint and Turán's theorem (Alon and Spencer, 2016, Theorem 3.2.1), each $\widetilde{I}_c$ contains an independent subset $I_c$ with $|I_c| = \Omega(|\widetilde{I}_c|)$. Since

$$|\widetilde{J}_c| = O(\delta(A_c) \log |V|) = O(|\widetilde{I}_c| \log^2 |V|) = O(|I_c| \log^2 |V|),$$

we conclude that $\sum_{c=1}^M |D_c| = O(\sum_{c=1}^M |I_c| \log^2 |V|) = O(\beta_M(G) \log^2 |V|)$.

### C.2 Proof of Lemma 3.2

The second inequality is straightforward: $\beta_{\text{dom}}(G, M) \leq \mathsf{m}(G)$ since $\cup_c B_c$ is acyclic, and the other inequality follows from $|B_c| = O(\delta(V_c) \log |V|) = O(\alpha(V_c) \log^2 |V|)$ in Lemma A.1. To

prove the first inequality, we consider a simple greedy algorithm for the learner, where $v_t \in A_{c_t}$ is the vertex with the largest out-degree in the induced subgraph by $A_{c_t} \backslash N_{\text{out}}(D_{t-1})$. Intuitively, $A_{c_t} \backslash N_{\text{out}}(D_{t-1})$ is the set of nodes in $A_{c_t}$ that remain unexplored by the learner by time $t$. Under this greedy algorithm, for $c \in [M]$, define

$$V_c = \bigcup_{t:c_t=c} \left( N_{\text{out}}(v_t) \bigcap (A_{c_t} \backslash N_{\text{out}}(D_{t-1})) \right), \qquad B_c = \{v_t : c_t = c\}.$$

We claim that $V_c$ are pairwise disjoint and $|B_c| \leq \delta(V_c)(1 + \log|V|)$, and thereby complete the proof of $\sum_{c=1}^{M} |B_c| \leq \beta_{\text{dom}}(G, M)$. The first claim simply follows from the pairwise disjointness of the sets $N_{\text{out}}(v_t) \bigcap (A_{c_t} \backslash N_{\text{out}}(D_{t-1}))$ for different $t$. For the second claim, let $t_1 < \cdots < t_n$ be all the time steps where $c_t = c$, and

$$V_{c,i} \triangleq \bigcup_{j=i}^{n} \left( N_{\text{out}}(v_{t_j}) \bigcap (A_c \backslash N_{\text{out}}(D_{t_j-1})) \right), \quad i \in [n].$$

It is clear that $V_{c,i+1} = V_{c,i} \backslash N_{\text{out}}(v_{t_i})$. Since $v_{t_i}$ has the largest out-degree in the induced subgraph by $A_c \backslash N_{\text{out}}(D_{t_i-1}) \supseteq V_{c,i}$, and $N_{\text{out}}(v_{t_i}) \cap (A_c \backslash N_{\text{out}}(D_{t_i-1})) = N_{\text{out}}(v_{t_i}) \cap V_{c,i}$, this is also the vertex with the largest out-degree in $V_{c,i}$. Therefore, the sets $\{V_{c,i}\}_{i=1}^{n+1}$ evolve from $V_{c,1} = V_c$ to $V_{c,n+1} = \varnothing$ as follows: one recursively picks the vertex with the largest out-degree in $V_{c,i}$, and removes its out-neighbors to get $V_{c,i+1}$. This is a well-known approximate algorithm for computing $\delta(V_c)$, described above Lemma A.2, with

$$|B_c| = n \leq \delta(V_c)(1 + \log|V_c|) \leq \delta(V_c)(1 + \log|V|),$$

as desired.

### C.3  Proof of Lemma 3.5

If $G$ is undirected, then $\beta_{\text{dom}}(G) \leq \mathsf{m}(G) = \alpha(G) = \beta_M(G)$ easily holds. It remains to consider the case where $G$ is transitively closed.

Note that in a transitively closed graph, every vertex set has an independent dominating subset (by tracing to the ancestors). Therefore, for the maximizing sets $B_1, \cdots, B_M$ in the definition of $\beta_{\text{dom}}$, we may run the above procedure to find independent dominating subsets $I_1, \cdots, I_M$ of $V_1, \ldots, V_M$ respectively, with $I_c \subseteq B_c$ and

$$\sum_{c=1}^{M} |I_c| \geq \sum_{c=1}^{M} \delta(V_c) \geq \frac{1}{C' \log|V|} \sum_{c=1}^{M} |B_c|.$$

Now consider the induced subgraph $G'$ by $\cup_c I_c$. Clearly $G'$ is acyclic, and the length of longest path in $G'$ is at most $M$ (otherwise, two points on a path belong to the same $I_c$, and transitivity will violate the independence). Invoking Lemma 4.1 now gives

$$\sum_{c=1}^{M} |I_c| = \mathsf{m}(G') \leq \frac{\rho(G')}{M} \beta_M(G') \leq \beta_M(G') \leq \beta_M(G),$$

and combining the above two inequalities completes the proof.

### C.4  Proof of Theorem 3.3

The proof is decomposed into three claims: with probability at least $1 - \delta$,

1. for every $c \in [M]$ and $\ell \in \mathbb{N}$, when the confidence bound (5) is formed, either every action in $A_{c,\ell}$ has been observed for at least $\ell$ times or $|A_{c,\ell}| = 1$;

2. for every $c \in [M]$ and $\ell \in \mathbb{N}$, the best action $a^\star(c)$ under context $c$ belongs to $A_{c,\ell}$, and every action in $A_{c,\ell}$ has suboptimality gap at most $\min\{1, 4\Delta_{\ell-1}\}$, with $\Delta_\ell \triangleq \sqrt{\log(2MKT/\delta)/\ell}$;

3. for every $\ell \in \mathbb{N}$, the total number $N_\ell$ of actions taken on layer $\ell$ in those subsets $A_{c,\ell}$ with $|A_{c,\ell}| > 1$ is $O(\beta_M(G) \log^2 K)$.

The first claim simply follows from (1) when $G$ is strongly observable, any subset of size $> 1$ is strongly observable, and (2) $A_{c,\ell} \subseteq N_{\text{out}}(\cup_{c' \leq c} D_{c',\ell})$ required by the sequential game I, so that on each layer $\ell' \in [\ell]$ there is at least one action which observes $a \in A_{c,\ell} \subseteq A_{c,\ell'}$. For the second claim, the first claim, the usual Hoeffding concentration, and a union bound imply that $|\bar{r}_{c,a} - \mu_{c,a}| \leq \Delta_\ell$ in (5) for all $c \in [M]$ and $a \in A_{c,\ell}$, when $|A_{c,\ell}| > 1$ and with probability at least $1 - \delta$. Conditioned on this event:

- The best arm $a^\star(c)$ is not eliminated by (5), because $\bar{r}_{c,a^\star(c)} \geq \mu_{c,a^\star(c)} - \Delta_\ell \geq \max_{a' \in A_{c,\ell}} \mu_{c,a'} - \Delta_\ell \geq \max_{a' \in A_{c,\ell}} \bar{r}_{c,a'} - 2\Delta_\ell$;

- The instantaneous regret of choosing any action $a \in A_{c,\ell+1}$ is at most $\min\{1, 4\Delta_\ell\}$, for $\mu_{c,a} \geq \bar{r}_{c,a} - \Delta_\ell \geq \bar{r}_{c,a^\star(c)} - 3\Delta_\ell \geq \mu_{c,a^\star(c)} - 4\Delta_\ell$, and $|\mu_{c,a} - \mu_{c,a^\star(c)}| \leq 1$ trivially holds.

Consequently the second claim holds for $|A_{c,\ell}| > 1$. For the case $|A_{c,\ell}| = 1$, note that starting from $A_{c,1} = [K]$, the above argument implies that the best arm is never eliminated until $|A_{c,\ell'}| = 1$ at some layer $\ell' \leq \ell$ conditioned on the high probability event, which implies the single action in $A_{c,\ell}$ is the best action and incurs 0 regret. The last claim is simply the reduction to the sequential game I, where Lemma 3.1 shows that $N_\ell = \sum_{c=1}^{M} |D_{c,\ell}| = O(\beta_M(G) \log^2 K)$.

Combining the above claims and that we incur 0 regret whenever $|A_{c,\ell}| = 1$, with probability at least $1 - \delta$, we have

$$R_T(\text{Alg 1}; G, M, \mathcal{C}_{\text{SA}}) \leq \sum_{\ell=1}^{\infty} N_\ell \min\{1, 4\Delta_\ell\},$$

where $N_\ell \leq N := O(\beta_M(G) \log^2 K)$, and $\sum_{\ell=1}^{\infty} N_\ell = T$. It is straightforward to see that the choice $N_1 = \cdots = N_m = N$ and $N_{m+1} = T - Nm$ for a suitable $m \in \mathbb{N}$ maximizes the above sum, and the maximum value is the target regret upper bound in Theorem 3.3.

## C.5 Proof of Theorem 3.4

The proof follows verbatim the same lines in the proof of Theorem 3.3, except that the total number $N_\ell$ of actions taken on layer $\ell$ is now at most $\beta_{\text{dom}}(G, M)$ in the third claim, by Lemma 3.2.

## C.6 Proof of Lemma 4.1

Let $V_1 \subseteq V$ be a maximum acyclic subset, then $\rho(G|_{V_1}) \leq \rho(G)$. Consider the following recursive process: at time $t = 1, 2, \cdots$, let $J_t$ be the set of vertices in $V_t$ with in-degree zero (which always exist as $V_t$ is acyclic), and $V_{t+1} = V_t \backslash J_t$. This recursion can only last for at most $\rho(G)$ steps, for otherwise there is a path of length larger than $\rho(G)$. Then each $J_t$ is an independent set, for every vertex of $J_t$ has in-degree zero in $V_t \supseteq J_t$. For the same reason we also have $J_i \not\rightarrow J_j$ for $i > j$. This means that

$$\mathsf{m}(G) = |V_1| = \sum_t |J_t| \leq \max\left\{\frac{\rho(G)}{M}, 1\right\} \beta_M(G),$$

where the last inequality follows from picking $M$ largest sets among $\{J_t\}$.

## C.7 Proof of the statement in Section 4.2.2

In this section, we show that when $G_{[K]}$ and $G_{[M]}$ are either both undirected or both transitively closed, the product graph quantities $\beta_{\text{dom}} := \beta_{\text{dom}}(G_{[K]} \times G_{[M]})$ and $\beta_M := \beta_M(G_{[K]} \times G_{[M]})$ satisfy

$$\beta_{\text{dom}} = O(\beta_M \log K).$$

Combining with the $\Omega(\sqrt{\beta_M T})$ lower bound, this shows the tightness of the upper bound $\widetilde{O}(\sqrt{\beta_{\text{dom}} T})$. The idea here is similar to Section C.3:

When $G_{[K]}$ and $G_{[M]}$ are both undirected, the union set $\cup_c B_c$ in the definition of $\beta_{\text{dom}}$ is an independent set thanks to the acyclic requirement. Thus $\beta_{\text{dom}} \leq \beta_M$.

When $G_{[K]}$ and $G_{[M]}$ are both transitively closed, for the maximizing sets $B_1, \ldots, B_M$ in the definition of $\beta_{\mathsf{dom}}$, we can again find independent dominating subsets $I_c \subseteq B_c$ (by transitive closure of $G_{[K]}$) with

$$\sum_{c=1}^{M} |I_c| \geq \frac{1}{C' \log K} \sum_{c=1}^{M} |B_c| = \frac{1}{C' \log K} \beta_{\mathsf{dom}}. \tag{9}$$

Now it suffices to find independent subsets $J_c \subseteq [K] \times \{c\}$ that satisfy $J_c \nrightarrow J_{c'}$ when $c < c'$ and $\sum_c |J_c| = \sum_c |I_c|$. Toward this end, we first suppose there are $c$ and $c'$ such that $u_c \to u_{c'}$ and $v_c \leftarrow v_{c'}$ for $u_c, v_c \in I_c$ and $u_{c'}, v_{c'} \in I_{c'}$. This implies $c \leftrightarrow c'$ in $G_{[M]}$ by the product graph structure. Then

- $I_c|_{[K]}$ and $I_{c'}|_{[K]}$ are disjoint since $\bigcup_c I_c$ is acyclic;
- by transitive closure of $G_{[K]}$ and that $I_c, I_{c'}$ are independent, $I_c|_{[K]} \cup I_{c'}|_{[K]}$ has path length at most 1;
- from above, there exist disjoint and independent sets $S_1$ and $S_2$ such that $S_1 \cup S_2 = I_c|_{[K]} \cup I_{c'}|_{[K]}$ and $S_1 \nrightarrow S_2$.

where we denote the set projection

$$S|_{[K]} = \{a \in [K] : (a, c) \in S \text{ for some } c \in [M]\}.$$

Without loss of generality, assume $c < c'$. In this case, we can "rearrange" them by letting $J_c = S_1 \times \{c\}$ and $J_{c'} = S_2 \times \{c'\}$, so $J_c \nrightarrow J_{c'}$. Now suppose there is a loop on the set level, i.e. there are $c_1, \ldots, c_m \in [M]$ with $I_{c_1} \to \cdots \to I_{c_m} \to I_{c_1}$. Similarly, we must have that $\{c_1, \ldots, c_m\}$ form a clique in $G_{[M]}$, $I_{c_1}|_{[K]}, \ldots, I_{c_m}|_{[K]}$ disjoint in $G_{[K]}$, and the path length in $\bigcup_{j=1}^{m} I_{c_j}$ is at most $m$. Then we can again "rearrange" them and get independent sets $J_{c_j} \nrightarrow J_{c_k}$ for $c_j < c_k$ and $j, k \in [m]$, and $\sum_{j=1}^{m} |J_{c_j}| = \sum_{j=1}^{m} |I_{c_j}|$. In other cases, we simply let $J_c = I_c$ and arrive at $J_1, \ldots, J_M$ that are acyclic on the set level, i.e. up to reordering of the indices, we have $J_c \nrightarrow J_{c'}$ when $c < c'$. Together with Eq (9), we have

$$\beta_{\mathsf{dom}} \leq C' \log K \sum_{c=1}^{M} |I_c| = C' \log K \sum_{c=1}^{M} |J_c| \leq C' \beta_M \log K.$$

