# OpenReview forum: "Stochastic contextual bandits with graph feedback: from independence number to MAS number"
_NeurIPS.cc/2024/Conference — NeurIPS 2024 poster_

### Official Review · Reviewer_DvmA · 2024-06-21

**Soundness:** 3
**Presentation:** 2
**Contribution:** 3
**Rating:** 6
**Confidence:** 4

**Summary:**

The authors consider the problem of contextual bandits with finitely many contexts, stochastic rewards and a directed feedback graph (assumed to contain all self loops) across actions. They study the setting of "complete cross-learning" where the reward feedback of the chosen action is observed across all contexts. In this setting, the authors establish a lower bound of $\Omega(\sqrt{\beta_M(G) T})$ where $\beta_M(G)$ is a graph-dependent quantity which lies between the independence number and the maximum acyclic subgraph of the feedback graph, and prove that this lower bound is tight for a specific class of context sequences by designing an efficient algorithm with a matching regret bound. Furthermore, the authors provide an upper bound for general context sequences with a graph dependence that improves upon the maximum acyclic subgraph, but in general may not be tight.

**Strengths:**

* The authors establish regret upper and lower bounds that constitute a considerable step towards characterizing the minimax regret in contextual bandits with feedback graphs in the complete cross-learning framework.
* The authors provide an overview of the hard instance construction which helps the reader understand the difficulty of minimizing regret in this setting.
* The idea of incorporating a sequential graph-theoretic zero sum game in a bandit algorithm seems novel and very interesting.
* Even though the authors close the gap completely only for specific types of context sequences, they also provide an upper bound for general sequences which improves upon the known upper bounds in the literature.

**Weaknesses:**

* My main issue with the presentation of the main results is with the comparison with the previous related work of [1] (lines 147-148). While the authors do mention that this work considers the setting of feedback graphs over the contexts as well as over the arms, it should be mentioned that it is not how the problem is presented in [1]. Rather, they consider a tabular RL problem in which the states correspond to the contexts, with the crucial difference that the transition between states is governed by some stochastic process. Since in the authors' work there is no assumption regarding the transition between contexts (that is, it could be adversarial), it seems not quite fair to compare their results to those of [1], specifically the regret lower bound. Indeed, the lower bound described by the authors is easily seen to be inapplicable in the setting of [1] as they vary the contexts (or states) in a very controlled adversarial manner. Therefore, I think the authors should emphasize that their lower bound does not provide a strengthening of the lower bound given in [1] for the RL setup.



References:
[1] Christoph Dann, Yishay Mansour, Mehryar Mohri, Ayush Sekhari, and Karthik Sridharan. Reinforcement learning with feedback graphs.

**Questions:**

I would appreciate it if the authors could address my main concern under "Weaknesses" and provide a more careful comparison with the work of [1].

**Limitations:**

Yes.

---

> ### Author Rebuttal · Authors · 2024-08-07
>
> We thank the reviewer for highlighting the difference between [1] and our work. Indeed [1] primarily studies the RL setting so their upper bound is more general than ours; however, our lower bound instance could be embedded into the tabular RL setting of [1]. Consider an episodic tabular RL with $H = 1$, so that the initial state at every episode is the context. Since the initial state can be adversarially chosen in [1], our lower bound construction is legitimate in this setting and could lead to better dependence on the graph structure. The general H can also be handled using an absorbing state. We will provide a more detailed comparison in the final version.

---

> > ### Comment · Reviewer_DvmA · 2024-08-10
> >
> > Thank you for your response. I indeed missed the fact that [1] considered a setting where the initial state can be adversarial, and thus includes the setting studied in this paper if $H=1$.
> >
> > I currently have no further questions, and will maintain my score.

---

### Official Review · Reviewer_iUw4 · 2024-06-26

**Soundness:** 3
**Presentation:** 3
**Contribution:** 3
**Rating:** 6
**Confidence:** 2

**Summary:**

In this paper, the authors consider the problem of contextual bandits with a feedback graph, for finite context space.  In the presented setting, taking an action reveals the rewards for all neighboring actions in the feedback graph for all contexts. The authors propose $\beta_M(G)$, a theoretical quantity in which $M$ is the number of contexts and $G$ is the feedback graph, with the goal of characterizing the hardness of learning for this class of problems. The authors prove a lower bound of  $\Omega(\sqrt{\beta_M(G)T})$ where $T$ is the number of rounds.  The authors also present a near-optimal upper bound of $\widetilde{O}(\sqrt{\min (\bar{\beta}(G),m(G))T})$.

**Strengths:**

1.	The insights in the paper are interesting, it characterizes the difference between MAB with graph feedback and Contextual MAB with graph feedback, for small and finite context space.

2.	The authors present upper and lower bounds for the problem.

3.	The algorithmic approach taken to produce both results is elegant, especially the use of the arm elimination technique.

**Weaknesses:**

1.	In contextual MAB literature, the main difficulty is that the context space is large, and in each episode, the feedback is revealed only for the current context. In the discussed setting, the feedback is revealed for all contexts. This significantly reduces the inherent difficulty of the contextual influence. Hence,

(a)	Can you please explain how having a context made the learning harder w.r.t the non-contextual MAB with feedback graph in this setting, conceptually?

(b)	Have you considered the standard Contextual setting in which only the feedback of the current context is released? Can you adjust your results to this setting as well?

2.	It would benefit the reader if an additional explanation of equation (4) will be given. So is for the result stated in Lemma 3.1.

3.	Typos - A comment that was left in the text: line 20: "to name a few".

**Questions:**

1.	See weakness 1.

2.	Can you please explain the conclusions to corollary 1.2? Specifically, the dependency of $I_c$ in the context is unclear to me, and why it implies that $\beta_M(G) = m(G)$.

3.	Can you please provide some more intuition regarding the behavior of $\beta_M(G)$, and an example for a calculation of it for simple graphs?

**Limitations:**

-

---

> ### Author Rebuttal · Authors · 2024-08-07
>
> We thank the reviewer for the review and the insightful questions.
>
> 1.(a). In general, the oracle we compare in contextual bandits to is able to take different optimal actions under different contexts, so it is a stronger benchmark. Also, our work assumes adversarial context, so intuitively, a larger number of contexts increases the adversary’s ability to break the learner’s exploration plans and to craft a “harder” problem instance. The latter corresponds to increasing $M$ in the lower bound quantity $\beta_M$ and thereby including extra independent subsets in the hard instance.
>
> 1.(b). Yes it is straightforward to extend to the case with no information across contexts (by running $M$ instances of the single-context algorithm), which essentially becomes $M$ separate subproblems and leads to an optimal regret $\widetilde{O}(\sqrt{M\alpha T})$.
>
> 1.(c). It is however an interesting question when the feedback across contexts is not complete. Section 4.1 discusses the product graph case for weakly observable action graphs. We define product graphs as $(a_1, c_1)\rightarrow (a_2,c_2)$ if $a_1\rightarrow a_2$ and $c_1\rightarrow c_2$ in their respective graphs (we'll make this definition clearer in text). When the action graph is strongly observable, we may extend our Theorem 1.1 and 1.3 with the following graph quantity (defined by both the action graph and the context graph) that shares the same spirit:
> $$
> \beta' = \max \lbrace\sum_{c=1}^{M}|I_c| : I_c\subseteq V_c \textnormal{ independent and } I_j\not\rightarrow I_k \textnormal{ if } j<k \rbrace
> $$
> and Theorem 1.4 with
> $$
> \bar\beta' = \max\lbrace \sum_{c=1}^{M}|I_c| : I_c\subseteq V_c \textnormal{ independent} \rbrace
> $$
> where $V_c=\lbrace (a,c): a\in[K] \rbrace$.
>
> 2. One can think of Eq (2) of $\beta_M$ as sequentially taking out one independent subset (and its out-neighbors) as the hard instance for a context. Note those $I_c$’s are allowed to be empty. By the constraint of $I_j \not\rightarrow I_k$ for $j<k$, they form a DAG on the set level, so their total size cannot exceed the MAS number $\mathsf{m}(G)$. Then when $M\ge \mathsf{m}(G)$, one can simply take the max acyclic subgraph and put one node for each $I_c$, while leaving $(M-\mathsf{m}(G))$ of them empty. This shows $\beta_M=\mathsf{m}(G)$ when $M$ is sufficiently large, hence Corollary 1.2.
>
> 3. As an example, consider a “monotone” graph across the actions: for actions in $[K]$, there is an edge $i\rightarrow j$ iff $i<j$. With cross-learning among $M$ context, the quantity $\beta_M=\max\lbrace M, \mathsf{m}(G)\rbrace$ increases from $\alpha=1$ to $\mathsf{m}(G)=K$ as $M$ grows. This example shows up in bidding in auctions (see Han et al. 2020 in our reference) with actions modeling the bidding values and contexts the values of the presented items.

---

> ### Comment · Reviewer_iUw4 · 2024-08-08
>
> I thank the authors for their response and have no further questions.

---

### Official Review · Reviewer_5kFj · 2024-07-10

**Soundness:** 3
**Presentation:** 3
**Contribution:** 3
**Rating:** 6
**Confidence:** 4

**Summary:**

This paper studies contextual online learning when the feedback received by the learner is regulated by a feedback graph. The setting is as follows: the actions constitute the nodes of a directed graph $G$, and playing action $a$ at time $t$ when the context is $x_t$ reveals not only the loss incurred by that action at that time, for that context but the losses of all the neighboring actions, for all possible $m$ contexts. While the contexts are generated adversarially, the losses are i.i.d..

The non-contextual problem is well understood, with tight minimax regret guarantees holding for both the adversarial and stochastic settings. These rates depend on both the time horizon $T$ and some graph parameters. In particular, for strongly observable graphs (as the ones studied in this paper), the rate is known to be $\sqrt{T \alpha}$, where $\alpha$ is the independence number of the feedback graph.

Previous results achieve a regret bound of $O(\sqrt{T m})$ for the contextual problem, where $m(G)$ is the maximum acyclic subgraph number, which is complemented by the abovementioned $\Omega(\sqrt{T \alpha})$ lower bound. This paper investigates the gap in the graph-dependent parameter in the minimax rate. Note, $\alpha$ = $m$ for undirected graphs.

The contribution of the paper are as follows:
when the number of contexts is large ($\ge m$), then the $\sqrt{T m}$ rate is tight
in general, a lower bound of $\Omega(\sqrt \beta_m)$ is proved, where $\beta_m$ is a new graph parameter which crucially depends on the number $m$ of contexts and gracefully interpolates between $\alpha$ and $m$.
improved upper bounds are then proved for special context sequences and the general problem

**Strengths:**

1. Online learning with feedback graphs is a relevant problem with a long literature in NeurIPS and ICML
2. Studying this problem with contexts is fairly natural, and has already been studied
3. the paper presents a consistent set of results and manages to present them nicely in the intro. Due to space constraints, the technical parts are, however, only roughly sketched.

**Weaknesses:**

- the problem is not closed: there is still a significant gap in the right graph theoretic parameter
- the result only holds in the stochastic setting. What can be said in the adversarial setting?
- the lower-bound construction is fairly natural (at a high level)
- the graph theoretic parameter introduced is artificial and way less natural than the ones present in the non-contextual settings.

Minor comments
- please update the references: e.g., Schneider and Zimmert and Zhang et al have been published
- please consider adding some further references to the learning with feedback graph literature. (see also questions)

**Questions:**

What is the relationship between your graph parameter and the ones in Eldowa et al ”On the Minimax Regret for Online Learning with Feedback Graphs”. NeurIPS 2023?

**Limitations:**

No potential negative societal impact

---

> ### Author Rebuttal · Authors · 2024-08-07
>
> We thank the reviewer for the review and the insightful questions.
> 1. Our work assumes adversarial context. When the reward is also adversarial, as shown in (Balseiro et al. 2019), cross-learning is not helpful and the optimal minimax regret is proved to be $\sqrt{M\alpha T}$: this essentially corresponds to dividing the horizon into $M$ different subproblems each with duration $T/M$. Then the minimax regret for each (single-context) subproblem is known to be $\sqrt{\alpha T/M}$.
> 2. (Eldowa et al. 2023) studies the non-context (or single-context) setting and arrives at the independence $\alpha$ number. Our proposed quantity $\beta_M$ can be seen as an extension of $\alpha$ under the assumption of adversarial context, as it sequentially takes the max independent subset and its neighbors. The information-flow constraint in Eq (2) then naturally arises from this adversarial assumption too.
> 3. We agree that the idea behind the lower bound construction is natural, but we’d also like to make two remarks. First, this natural construction proves the tightness of the MAS number when the context space is large, an observation overlooked in the literature even when the upper bound using the MAS number was obtained. Second, the proof technique requires to achieve a careful balance between exploration and exploitation, and the two-inequality approach (below Line 182) could be of general interest when proving interactive lower bounds.
> 4. Although our graph-theoretic quantity $\beta_M$ is not as natural as the independence number and the MAS number, our lower bound shows that this is the right quantity for contextual bandits (at least under self-avoiding contexts). In addition, this quantity exhibits the desired interpolation between the independence number and MAS number.
> 5. We appreciate the reviewer for mentioning these references and shall include them and further discussion in the paper.

---

### Official Review · Reviewer_mcAY · 2024-07-13

**Soundness:** 3
**Presentation:** 3
**Contribution:** 3
**Rating:** 6
**Confidence:** 3

**Summary:**

In this work, the authors consider the problem of contextual bandits with feedback graphs and aim to achieve a tighter dependency on graph-dependent quantities.
Figuring out the correct dependency on graph-dependent quantities is a notably challenging problem in the standard MAB framework, as aspects such as whether the feedback graph is directed, whether it has self-loops, or whether the time horizon is large compared to the size of the graphs are crucial components to figure out how much information can be extracted from the feedback graph.\\

In this work, the authors consider a contextual MAB problem where there is a feedback graph across actions and cross-learning between contexts. The authors propose a minimax lower bound for this context, which depends on a quantity $\beta_M(G)$, where $M$ is the number of contexts.
The authors then show that this lower bound is tight for certain classes of problems such as self-avoiding contexts, which is a problem setting where the environment regularly switches from one context to the next but doesn't ever come back to contexts that have already been seen in the past.
The authors also derive an upper bound in the general setting using a different algorithm.

**Strengths:**

The authors propose a detailed characterization of the challenges of contextual MAB with feedback graphs by focusing in deriving a lower bound and gaining a good sense of how challenging the problem is. They then provide both an algorithm in a setting where they can achieve a tight bound as well as a general algorithm.

The proofs are well detailed, and the authors properly study and discuss the gap between upper and lower bounds as well as possible extensions.

**Weaknesses:**

While the gap in terms of upper and lower bounds may be tight in terms of the graph-dependent quantities in settings with self-avoidant contexts, the upper bounds contain some superfluous logarithmic dependencies. (In particular the $log^2 K$ term in Theorem 3.3). Do you think that this could be avoided?

**Questions:**

See weaknesses.

You propose a specific algorithm for the setting with self-avoidant contexts, but do you think it is a realistic assumption to make ahead of time?
Do you think that it would be possible to get some sort of best-of-both-worlds guarantees, where the same algorithm could achieve tight bounds if we are in the self-avoidant setting while still ensuring worst-case guarantees in the general case?

**Limitations:**

Theoretical work, NA

---

> ### Author Rebuttal · Authors · 2024-08-07
>
> We thank the reviewer for the review and the insightful questions.
> 1. While we agree it is possible to improve the logarithmic $\log(MKT)$ term with a more careful concentration argument, it is less obvious how to avoid the $\log^2K$ term when we use a practical algorithm: if we are allowed access to the dominating subset of the active set of arms, as mentioned in Lemma 3.1, we are only left with $\log K$ term. This term comes from Lemma A.1 and that we include both dominating subsets and independence subsets in the subroutine of Algorithm 1 in order to relate our exploration sets to independent sets. We are not sure this can be removed in our approach.
> 2. While we include it more as a mathematically interesting case, we wish to point out that our bounds are also tight when the graph is undirected or transitively closed (Theorem 1.4), which is an often more realistic assumption. It is achieved by the more straightforward Algorithm 2 which just greedily picks the active arm with most out-going edges in the subroutine. As an example, in bidding in auctions where the winning bid is revealed, the feedback graph is transitively closed across the learner’s actions (e.g. Han et al. 2020 in our reference).
> 3. This is a great question. In our Example 1, while the quantity $\beta_{dom}$ used in bounding Algorithm 2 (Lemma 3.2) is loose, Algorithm 2 itself actually achieves the optimal regret under appropriate tie-breaking strategy. It is open whether we can arrive at a tighter bound for Algorithm 2 directly or with modifications, such as exploring an extra small independence subset as we did in Algorithm 1 (details in Appendix C.1).

---

> > ### Comment · Reviewer_mcAY · 2024-08-07
> >
> > Thank you for the extra clarifications. I currently don't have further questions.

---

### Official Review · Reviewer_6nrr · 2024-07-26

**Soundness:** 3
**Presentation:** 3
**Contribution:** 2
**Rating:** 5
**Confidence:** 3

**Summary:**

This paper investigates the problem of stochastic contextual bandits with graph feedback, in which a graph over actions models the feedback structure. The learner selects an action after observing the current context, and then receives the losses of the actions that are neighbors of the selected one in the feedback graph.
This work proposes a novel graph-theoretic quantity $\\beta\_{M}(G)$ to characterize the statistical complexity of learning in this problem setting.
The authors establish a minimax regret lower bound $\\Omega(\\sqrt{\\beta\_{M}(G)T})$, where $\\beta\_{M}(G)$ interpolates between the independence number $\\alpha(G)$ and the maximum acyclic subgraph (MAS) number $m(G)$.
Specifically, $\\beta\_{M}(G) = \\max \\{\\sum\_{c=1}^{M} |I\_c| : I\_1, \\dots, I\_M \\text{ are independent sets in } G, I\_i \\nrightarrow I\_j \\text{ for } i < j\\}$.
This result implies that, while $\\alpha(G)$ dictates the complexity in multi-armed bandits (i.e., $M=1$), $m(G)$ becomes the relevant parameter as the number of contexts increases.
The paper further provides algorithms that achieve near-optimal regret bounds $\\tilde{O}(\\sqrt{\\beta\_{M}(G)T})$ for self-avoiding context sequences and $\\tilde{O}(\\sqrt{\\min\\{m(G),\\bar\\beta\_{M}(G)\\} T})$ for general context sequences (where $\\bar\\beta\_{M}(G)$ is a larger but similarly defined graph parameter than $\\beta(G)$), leveraging carefully designed arm elimination techniques.
These algorithms are polynomial-time and demonstrate tight regret bounds for special families of context sequences and feedback graphs, namely undirected or transitively closed graphs.

**Strengths:**

The most interesting contribution of this work is probably the connection between the learnability of the problem and the novel (at least to the best of my knowledge) graph-theoretic parameter $\\beta_{M}(G)$.
Thanks to this, the authors are able to provide further insights into the contextual bandit problem with feedback graphs (with complete cross-learning and under the assumption), showing that the number of contexts can influence the dependence of the regret on the structure of the feedback graph $G$, interpolating between the independence number $\\alpha(G)$ (when $M=1$) and $m(G)$ (e.g., when $M \\ge m(G)$.
The way the regret analysis shows the dependence on such a parameter, via the computation of the value of the sequential game described in Section 3, is also interesting and nontrivial.

**Weaknesses:**

What the authors consider in this work is not the entire family of strongly observable feedback graphs (the one known to correspond with minimax regret of order $\\sqrt{T}$ in the non-contextual case), but only a subset of those graphs, i.e., that contain all self-loops. This excludes relevant feedback graphs such as the loopless clique and the apple tasting one. I think a discussion about these missing graphs, e.g., in Section 4.1 would give a clearer picture of the contributions of this work and how they compare with previous relevant work.

More importantly, the only nearly tight bounds for the regret are provided for self-avoiding contexts.
Moreover, the setting of complete cross-learning studied in this work seems quite restrictive as it imposes the assumption of observing the reward of the chosen action under any context.
This feels like a somewhat limiting assumption, as in real-world scenarios such a reward is often observed for the current context only, and the same context could reappear in non-contiguous time steps.
The applicability of the results is nevertheless sufficient, especially given some applications of interest and the extension of their results for the more general setting, albeit lacking a nearly optimal characterization for general context sequences and any strongly observable feedback graphs.

A further limitation of the applicability of the results is the fact that the feedback graph is assumed to be fixed.
This might not be the case generally speaking, as feedback graphs could be time-varying (as assumed in most of the recent literature on bandits with feedback graphs).
Time-varying feedback graphs could also be found in applications such as repeated first-price auctions (e.g., see “The Role of Transparency in Repeated First-Price Auctions with Unknown Valuations” by Cesa-Bianchi, Cesari, Colomboni, Fusco, and Leonardi, STOC 2024), which is one of the applications mentioned in the related work within this paper.

**Questions:**

- Please, address any relevant doubt that might have arisen from what is written above.
- Can the results be extended to the other strongly observable feedback graphs (i.e., the ones not containing all self-loops)? What could be the technical limitations, if you see any?
- Do you think it could be possible to adapt the algorithmic techniques in this work for time-varying graphs?

Minor comments/typos:
- Throughout the paper, use “domination number” instead of “dominating number”
- Line 84: “In what follows” instead of “In the sequel”
- Some references mention the arXiv, while they might already be published in some conference or journal
- I think the introduction would benefit from a more thorough comparison with the literature on bandits with feedback graphs. For instance, detailed studies on the minimax regret for bandits with feedback graphs have been pursued in:
  - Chen, Huang, Li, and Zhang. “Understanding bandits with graph feedback”, NeurIPS 2021
  - Eldowa, Esposito, Cesari, and Cesa-Bianchi. “On the minimax regret for online learning with feedback graphs”, NeurIPS 2023
  - Chen, He, and Zhang. “On interpolating experts and multi-armed bandits”, ICML 2024
- Lines 157-159: the following paper also fits with that description:
  - Zhang, Zhang, Luo, and Mineiro. “Efficient contextual bandits with uninformed feedback graphs”, ICML 2024

---

> ### Author Rebuttal · Authors · 2024-08-07
>
> We thank the reviewer for the thorough review and insightful questions.
> 1. It is indeed a great question about extending to the class of all strongly observable graphs, and it turns out that the extension is straightforward.
>
> Upper bounds: A key implication of strongly observable graphs is that for any subgraph with size > 1, there is a dominating subgraph. In Sec. 3 we may safely assume the subgraphs are larger than 1 because having a singleton active subset in arm elimination means it is the optimal arm with high probability. In addition, Lemma A.1 and A.2 still hold for strongly observable graphs. Therefore, the minimax quantities in Section 3 remain to hold.
>
> Lower bound: The same lower bound still holds, after adding an additional requirement that $I_1, …, I_M$ are disjoint in the definition of $\beta_M(G)$ in (2). In fact, for strongly observable nodes with no self loop, they cannot appear in $I_2, …, I_M$, and if they appear in $I_1$ we must have $|I_1| = 1$. Therefore, our lower bound analysis still works through.
>
> 2. We agree that the assumption of self-avoiding context is often restrictive in applications, and we include it as a mathematically interesting case. However, we wish to point out that our bound is also tight under the assumption of undirected or transitively closed graphs (Theorem 1.4) with the easier-to-implement Algorithm 2. We believe this assumption finds a wider range of applications, such as bidding in auctions where the feedback graphs are transitively closed (e.g. Han et al. 2020) and also possess complete cross-learning.
>
> 3. We agree that an incomplete cross-learning is an interesting open question, and discuss it partially in Section 4.1 with weakly observable context-action product graphs. We define product graphs as $(a_1, c_1)\rightarrow (a_2,c_2)$ if $a_1\rightarrow a_2$ and $c_1\rightarrow c_2$ in their respective graphs (we'll make this definition clearer in text). When the action graph is strongly observable, we may extend our Theorem 1.1 and 1.3 with the following graph quantity (defined by both the action graph and the context graph) that shares the same spirit:
> $$
> \beta' = \max \lbrace\sum_{c=1}^{M}|I_c| : I_c\subseteq V_c \textnormal{ independent and } I_j\not\rightarrow I_k \textnormal{ if } j<k \rbrace
> $$
> and Theorem 1.4 with
> $$
> \bar\beta' = \max\lbrace \sum_{c=1}^{M}|I_c| : I_c\subseteq V_c \textnormal{ independent} \rbrace
> $$
> where $V_c=\lbrace (a,c): a\in[K] \rbrace$.
> 4. We agree and appreciate the reviewer for pointing out our inapplicability to time-varying graphs. The high-level reason is that although the graph feedback is helpful for multi-armed bandits even in the adversarial setting, it is typically not helpful for adversarial contextual bandits. For example, it was shown in (Balserio et al. 2019) that when the rewards are adversarial, even with cross learning it is optimal to handle each context separately, with the optimal regret scaling with the number of contexts. Therefore, while previous literature handles time-varying graphs in an adversarial framework using EXP3-type algorithms, we do not know of the counterpart for arm-elimination-based algorithms.
> 5. We will fix the typos and update the references pointed out by the reviewer.

---

> > ### Comment · Reviewer_6nrr · 2024-08-14
> >
> > I thank the authors for addressing the points raised in my review. I am currently keeping my score and will make a final decision after further discussing with the other reviewers and the AC.

---

### Author Rebuttal · Authors · 2024-08-07

We appreciate the insightful reviews and questions from the reviewers and would like to highlight points that have drawn most attention here.
1. In addition to self-avoiding context (Theorem 1.3), Theorem 1.4 shows that our results are also tight for arbitrary context when the feedback graphs are either undirected or transitively closed (i.e. extra assumptions on the information structure between actions rather than context). The latter appears to find more realistic applications including bidding in auctions and inventory control.
2. While we assume self-loops in the current work, as pointed out by one reviewer, our results actually apply to all strongly observable graphs. The only difference is to add an extra constraint that $I_1,...,I_M$ are disjoint in definition (2) of $\beta_M$.
3. It is an interesting question to look beyond complete cross-learning over contexts. We partially discuss this in Section 4.1 on context-action product graphs when the action graph is weakly observable, where product graphs are defined as $(a_1,c_1)\rightarrow (a_2,c_2)$ if $a_1\rightarrow a_2$ and $c_1\rightarrow c_2$ in their respective graphs. It is straightforward to extend our results to the case of product graphs when the action graph is strongly observable too.

---

### Decision · Program_Chairs · 2024-09-25

**Decision:**

Accept (poster)

**Comment:**

Reviewers find that the new graph theoretic quantity $\beta_M(G)$ to be interesting and are happy with how it is able to interpolate between the independence number and the MAS number of $G$ depending on the number of contexts. Reviewers are also happy with the fact that in  the setting of self-avoiding contexts the bounds appear nearly tight for most graphs. Reviewer 6nrr, however, correctly notes that the current bounds hold only for strongly observable graphs that have all self-loops and not for the entire set of strongly observable graphs. Further, multiple reviewers have expressed concerns about how realistic the self-avoiding contexts setting is. Reviewer 6nrr has also expressed concerns about the fact that the proposed algorithms will only work in the setting of a fixed feedback graph and can not easily be extended to time varying feedback graphs. However, all reviewers agree that the current work advances the study on what is the right graph theoretic quantity to characterize the regret in the stochastic CMAB problem with graph feedback.